# Who We Seek and What We Eat? Sources of Food Choice Inspirations and Their Associations with Adult Dietary Patterns before and during the COVID-19 Lockdown in New Zealand

**DOI:** 10.3390/nu13113917

**Published:** 2021-11-01

**Authors:** Rajshri Roy, Teresa Gontijo de Castro, Jillian Haszard, Victoria Egli, Lisa Te Morenga, Lauranna Teunissen, Paulien Decorte, Isabelle Cuykx, Charlotte De Backer, Sarah Gerritsen

**Affiliations:** 1Discipline of Nutrition and Dietetics, University of Auckland, Auckland 1023, New Zealand; t.castro@auckland.ac.nz; 2Department of Epidemiology and Biostatistics, University of Auckland, Auckland 1023, New Zealand; 3Department of Human Nutrition, University of Otago, Dunedin 9016, New Zealand; jill.haszard@otago.ac.nz; 4School of Nursing, University of Auckland, Auckland 1023, New Zealand; v.egli@auckland.ac.nz; 5Centre for Hauora and Health, Massey University, Palmerston North 4442, New Zealand; L.TeMorenga@massey.ac.nz; 6Department of Communication Studies, University of Antwerp, 2000 Antwerpen, Belgium; Lauranna.Teunissen@uantwerpen.be (L.T.); paulien.decorte@uantwerpen.be (P.D.); Isabelle.Cuykx@uantwerpen.be (I.C.); charlotte.debacker@uantwerpen.be (C.D.B.); 7School of Population Health, University of Auckland, Auckland 1023, New Zealand; s.gerritsen@auckland.ac.nz

**Keywords:** COVID-19, nutrition, dietary patterns, feeding behaviours, food influence, food preferences, cooking, surveys and questionnaires, social media, New Zealand

## Abstract

Research shows the shaping of food choices often occurs at home, with the family widely recognised as significant in food decisions. However, in this digital age, our eating habits and decision-making processes are also determined by smartphone apps, celebrity chefs, and social media. The ‘COVID Kai Survey’ online questionnaire assessed cooking and shopping behaviours among New Zealanders during the 2020 COVID-19 ‘lockdown’ using a cross-sectional study design. This paper examines how sources of food choice inspirations (cooking-related advice and the reasons for recipe selection) are related to dietary patterns before and during the lockdown. Of the 2977 participants, those influenced by nutrition and health experts (50.9% before; 53.9% during the lockdown) scored higher for the healthy dietary pattern. Participants influenced by family and friends (35% before; 29% during the lockdown) had significantly higher scores for the healthy and the meat dietary patterns, whereas participants influenced by celebrity cooks (3.8% before; 5.2% during the lockdown) had significantly higher scores in the meat dietary pattern. There was no evidence that associations differed before and during the lockdown. The lockdown was related to modified food choice inspiration sources, notably an increase in ‘comforting’ recipes as a reason for recipe selection (75.8%), associated with higher scoring in the unhealthy dietary pattern during the lockdown. The lockdown in New Zealand saw an average decrease in nutritional quality of diets in the ‘COVID Kai Survey’, which could be partly explained by changes in food choice inspiration sources.

## 1. Introduction

Optimal nutrition is important for improved health and well-being and reduces the risk of diet-related health conditions, including chronic disease. In Aotearoa, New Zealand, the Ministry of Health publishes the Eating and Activity Guidelines for healthy eating based on the best available scientific evidence, interpreted for the local population [1]. In recent years, the world of food has been quietly overtaken by an array of electronic devices, online content, and information and communication technologies [2]. People receive nutrition information through both media and interpersonal sources, because they are passively exposed to it through routine media use and conversation with others, or they actively seek such information [3,4]. The accessibility to nutrition information is now near-universal due to internet access, and the information available varies in its scientific integrity and provider expertise [5]. Various studies have shown that dietary advice could help populations with their daily meal choice and preparation, but few studies have a comprehensive overview of the type and frequency of food advice that is considered beneficial [6].

Nationally credentialed nutrition professionals such as dietitians, nutritionists, and scientists are best able to communicate sound advice and scientific advances about nutrition [7]. However, consumers do not always seek food-related advice from these experts [8]. People tend to believe information from a wide range of sources or “influencers”, particularly that which is reinforced by sports figures, celebrities, health food store personnel, teachers, coaches, ministers, legislators, media commentators, health professionals, and other persons they respect in life or online on social media [8]. These influencers are recognised as being significant in food decisions [9]. However, the internet has no filters on the quality or accuracy of the information, enabling myths and pseudoscience to proliferate rapidly. People declaring themselves as ‘subject experts’ who lack formal training, reputable credentials, or adherence to a professional code of conduct have a voice. Blogs, social media posts, and ‘expert’ websites can be written by anyone, regardless of expertise [5,10]. Additionally, often, social media influencers are selling something, and it is not always clear if the aim of the ‘advice’ is for personal profit [11]. Regardless of credentials or expertise, anyone can disseminate nutrition information online, putting the public at risk of receiving unreliable and harmful advice [12]. There is lack of research on who New Zealanders seek out for food- and health-related information and how this impacts their dietary patterns.

Additionally, the rise in convenience products and increases in eating food purchased away from home appears to parallel a decline in dietary quality, leading some to suggest that a growing cohort of individuals lack the necessary cooking skills and food preparation knowledge to allow for the production of healthy, home-cooked meals [13]. It is argued that people cannot be expected to consume food recommended in dietary guidelines if they do not know how to prepare the food [13,14]. People who cook food at home prefer to utilise social media to meet their cooking needs [15]. However, there is a lack of research that focuses on New Zealanders’ principal sources of recipes and the general propensity to use recipes from such sources [16]. There is a need for research into food and cooking practices, particularly in light of repeated COVID-19 pandemic lockdowns when certain foods may be in short supply; people had more time to cook during the lockdowns, as they were at home and convenience, restaurant, and takeaway foods were restricted [17].

In early 2020, as countries were taking more strict lockdown measures to contain the spread of the COVID-19 pandemic, self-quarantine and the temporary closing of businesses affected regular food-related practices. In New Zealand, strict lockdown included two alert levels, level 4 and level 3. During alert level 4, small grocers (fruit and vegetable shops), restaurants, and takeaway shops were closed, and some fresh items such as fruits, vegetables, whole grains, flour, and bread became less available. During alert level 3, takeaway shops, restaurants, and cafes reopened with no physical interaction with customers (contactless pick up or delivery). Supermarkets were limited to the number of people they could accommodate simultaneously, meaning that people had to queue outside the store and keep 2 m away from others during these alert levels. The stress of being in a lockdown caused by a global pandemic, significant loss of income [18], and limited access to fresh food could also have led to increased consumption of highly processed foods, which tend to be high in fats, added sugars, and salt [19]. At the same time, increased social media marketing and ‘COVID-washing’ was used by unhealthy food and drinks brands to increase brand loyalty and encourage consumption [20]. Nonetheless, being forced to stay home may have created an opportunity to focus on food and diet and increased time for access to food-related advice online or from family and friends, to continue eating a diet that supports good health [21].

It is important to understand which food-related advice and inspiration sources are associated with dietary patterns to identify areas of concern. Reasons for recipe selection possess a social significance that merits greater attention, for it is the starting point of many food choices and related activities [16]. Furthermore, the COVID-19 lockdown provides an opportunity to consider how sources of advice influence dietary patterns, as new routines and isolation have changed how we live our lives. In this study we: (1) described changes to the inspiration sources of food- and cooking-related advice sought and the reasons for recipe selection before and during the COVID-19 lockdown in New Zealand and (2) examined associations between dietary patterns [19], the sources of food inspiration and cooking-related advice, and the reasons for recipe selection before and during the lockdown.

## 2. Materials and Methods

### 2.1. The COVID Kai Survey and the Number of Participants in This Study

The COVID Kai Survey [19] is the New Zealand arm of the Corona Cooking Survey, an online international survey conducted in 38 countries that aimed to examine changes in planning, selecting, and preparing healthy foods in relation to personal factors (time, money, stress) and social distancing policies during the COVID-19 crisis [22]. The study’s questionnaire was developed by researchers from the University of Antwerp, Belgium, and loaded onto the Qualtrics survey platform for each participating country separately. Recruitment of participants was through convenience/snowball sampling [19]. Respondents were adults 18 years or older, with no further restrictions on participation. We aimed to recruit as widely as possible in the adult population and monitored responses by demographic groups of interest at several points during data collection. The survey was self-reported in a cross-sectional study design assessing both before and during the lockdown at the same time. The survey closed with *n* = 4104 entries. Spam responses (2%) indicated by the software programme were removed (*n* = 81), along with two respondents who gave implausible ages (as the rest of their survey answers may have been unreliable), and those that were resident outside Aotearoa New Zealand (*n* = 28, 0.7%). Those that did not complete the relevant questions from the survey (24%) were excluded from the analyses in this paper (*n* = 965) [17]. We further excluded 51 (1.2%) participants who had complete dietary information for only before or during the lockdown. Thus, the number of participants included in the present study was 2977 (Appendix A).

This study used participants’ information on sociodemographic characteristics (gender, ethnicity, level of education, and whether the household participants had any children), sources for food inspirations and cooking-related advice sought before and during the lockdown, reasons for recipes selection, and previously derived dietary patterns, described in detail below [15].

### 2.2. Sources for Food- and Cooking-Related Advice and Reasons for Recipe Selection

Participants could indicate more than one source of advice for the questions on types and frequency of sources of food- and cooking-related advice sought before and during the lockdown. The following questions were asked to participants ‘When you do grocery shopping, you make choices, and these choices can also be influenced by others (indirectly). How often did the following people/sources influence your usual food choices when you went grocery shopping?’ ‘Whose advice about the sorts of food you should eat for your health did you listen to? Before the lockdown and during the lockdown’. The options of sources were household members, family, friends and acquaintances, nutrition experts (dietitians or nutritionists), health experts (medical doctors), scientists, celebrity chefs, celebrities who are not professional cooks, food influencers (not chefs either, people that generate food content via any medium to reach and engage with their followers), and other people. For each of the inspiration sources, the options for frequency of influence were never; very rarely; rarely; sometimes; frequently; very frequently; every time. For analysis purposes, these questions were combined into four subscales: family and friends (8 items); nutrition and health experts (3 items); celebrity cooks (6 items); food influencers (3 items). Cronbach’s alpha showed good internal consistency for each of these scales before and during the lockdown (ranging from 0.77 to 0.85). As some of the data were right skewed, the subscales were dichotomised into ‘not influenced’ (never, very rarely, rarely) and ‘influenced’ (sometimes, frequently, very frequently, and every time).

Two questions assessed the participants’ reasons for selecting recipes before and during the lockdown. The first question was ‘There are many recipes and food preparation messages available to us. How often did you actively search for recipes/food preparation content?’ The options for this question were: never, very rarely, rarely, sometimes, frequently, very frequently, and all the time. The second question asked was: ‘When you select recipes, which of the following aspects do you take into consideration?’. The options for the aspects included: guaranteed to taste good, achievable with few ingredients, achievable with the ingredients that I have at home, achievable with ingredients that can be easily found at the store, easy to prepare, quick to prepare, innovative (new, something different), inexpensive to make, comforting, healthy, and environmentally friendly. For each of the aspirations, the response options were strongly disagree, disagree, somewhat disagree, neither agree nor disagree, somewhat agree, agree, or strongly agree. The variable reasons for recipe selection were derived only for participants who indicated that they actively searched for recipes/food preparation content at least ‘sometimes’. Among those, participants who indicated either ‘somewhat agree’, ‘agree’, or ‘strongly agree’ were classified as selecting recipes for the reason in question. Participants who indicated either strongly disagree, disagree, somewhat disagree, or neither agree nor disagree were classified as not selecting recipes for the asked reason.

Finally, respondents were asked in an open question to write down the name of their main food-related influential figure, organization, or brand whose recipes they used the most. Next, they were asked to rate this specific influencer on four source credibility items, trustworthiness, expertise, how relatable he/she/it is, and attractiveness [23] (Appendix A).

### 2.3. Dietary Patterns before and during the Lockdown

The dietary pattern analysis has been described in the main study publication and is not a finding of this current study [19]. Before and during the lockdown, the frequency of intake of 19 selected foods, drinks, and food groups were assessed using a food frequency questionnaire (FFQ). Participants were asked to report how often they had at least one portion/serving of the different items within a scale ranging from (almost) never to twice or more times a day [19]. As previously described in the original publication [19], the FFQ items were converted to daily equivalents (servings/day), and dietary patterns (DPs) were determined using principal component analysis (PCA). Four DPs were indicated before and during the lockdown using a scree plot. PCA was conducted separately for each time point to assess whether similar DPs were present before and during the lockdown, and the loading matrix was orthogonally rotated. Though there were small differences in loadings between pre-lockdown and lockdown data, the DPs identified were essentially the same in both time points [19]. A factor loading ≥0.30 or ≤−0.30 was considered significant for this sample size. The following DPs were identified before and during the lockdown: healthy—positive high loadings of intake for fruit, vegetables, legumes/pulses, nuts or nut spread, and whole meal breads/pasta/grains; unhealthy—positive high loadings for processed meat, sweet and salty snacks, white breads/pasta/grains, and sugared beverages; meat—positive high loadings unprocessed fish, poultry, and red meat; vegan—positive high loadings for vegetarian alternatives and plant-based drinks and negative loadings for milk and other dairy products. Together, these four DPs explained 42.9% and 41.8% of the variance in food intake, respectively, before and during the lockdown. The variance explained within each of the four patterns ranged from 12.8% (DP healthy before the lockdown) to 9.0% (DP meat before and during the lockdown) [19]. Scores for the dietary patterns were calculated by multiplying the loading from the pre-lockdown PCA by the participant response to each FFQ item (in serves/day) and summing. This was carried out for both pre-and during the lockdown FFQ data.

### 2.4. Statistical Analysis

All statistical analyses were undertaken in Stata 16.1 (StataCorp, College Station, TX, USA). Analyses describing sources of food- and cooking-related advice and their associations with dietary patterns included only participants who had completed the questions on sources of inspiration sources before and during the lockdown. The numbers and proportions of those who were ‘influenced’ by family and friends, nutrition and health experts, celebrity cooks, and food influencers were calculated separately for before and during the lockdown (presented for the whole sample and for male and female subgroups). The proportions of participants who ‘became influenced’ (i.e., were not influenced before the lockdown but were influenced during the lockdown) and the proportions who ‘dropped influence’ (i.e., were influenced before the lockdown but not influenced during the lockdown) were calculated to determine if certain inspiration sources increased or decreased during the lockdown.

Analyses describing participants’ reasons for recipe selection and their associations with dietary patterns included only participants classified as those who actively searched for recipes/food preparation and answered the questions of reasons for recipe selection, before and during the lockdown.

Linear regression models were used to determine associations between sources of food- and cooking-related inspiration sources and dietary patterns. Each model contained one of the identified dietary patterns (as the dependent variable) and one of the sources of influence as the independent variable. All models were adjusted by the participants’ age, gender, ethnicity, level of education, and whether the household participants had any children. To further examine if these relationships were different during the lockdown than before the lockdown, a mixed-effects regression model was run as for the linear models but with the participant as a random effect and an interaction term between the source of influence and lockdown status. The same analyses were undertaken for examining the associations between participants’ reasons for recipe selection and their scores in each of the four identified dietary patterns. Residuals of all models were plotted and visually assessed for heteroskedasticity and normality. Two-sided statistical significance was determined at *p* < 0.05.

## 3. Results

### 3.1. Characteristics of the Participants

The mean age of the sample was 44.3 years (SD: 14.0), and the age range was 18–87 years. The largest proportion (47.8%) was aged 30 to 49 years. Most respondents were female (88.6%), with a tertiary education qualification (76%), and identified as New Zealand European ethnicity (82.8%). One-third of the surveys (*n* = 1012) were completed under COVID-19 alert level 4 and 66.6% under alert level 3 (*n* = 2016). Time since the start of lockdown ranged from 31 days to 51 days (median: 40 days) (Gerritsen et al.) [19] .

### 3.2. Sources Sought of Food- and Cooking-Related Advice

Table 1 presents the number and percentage of participants according to their reported food and cooking inspiration sources before and during the lockdown. Half of the participants (50.9%) reported being influenced by nutrition and health experts for food and cooking before the lockdown. Nutrition and health experts dropped in influence during the lockdown for 15% of the participants and increased in influence for only 3% of the participants. Similar proportions of males and females reported that they were influenced by nutrition and health experts during the lockdown (3.2% and 3.5%, respectively), but higher proportion of females (16%) had been influenced by nutrition and health experts before the lockdown when compared to males (9%). Thirty-five percent of the participants reported being influenced by friends and family before the lockdown, and this proportion dropped to about 29% during the lockdown. The respective proportions of females and males who reported becoming influenced (4.6 and 4.2%) and dropping influence (11.5% and 10.9%) by friends and family during the lockdown were similar. The lowest reported category of inspiration source was celebrity cooks, where only 3.8% reported being influenced by them before the lockdown and 5.2% during the lockdown. Nadia Lim (television celebrity chef and dietitian) was named as the top food-related influential figure, organization, or brand in New Zealand whose recipes were used (35%), and it is because most of these respondents (56%) found her trustworthy.

### 3.3. Reasons for Recipes Selection

Table 2 presents the reasons for recipes selection before and during the lockdown. In total, 68.1% (*n* = 2063) of the participants answered the recipes questions about before the lockdown, and 73.2% (*n* = 2218) of them answered these questions during the lockdown. Both before and during the lockdown, the most frequently reported reasons for recipe selection were ‘guaranteed to taste good’ (88.0% before and 90.9% during the lockdown) and ‘achievable with ingredients that can be easily found at the store’ (88.0% before and 88.1% during the lockdown). The reason of recipe selection because it is ‘healthy’ was the third most common reported reason, before (85.5%) and during the lockdown (80.1%). Before the lockdown, the least frequently reported reason for selecting a recipe was ‘innovative (new, something different)’ (53%), and during the lockdown, the least frequently reported reason for selecting a recipe was ‘quick to prepare’ (51%).

### 3.4. Inspiration Sources of Food- and Cooking-Related Advice and Dietary Patterns before and during the Lockdown

Table 3 presents the results of the adjusted associations between food- and cooking-related inspiration sources reported by participants and their scores in the same dietary patterns before and during the lockdown. Participants who reported being influenced by nutrition and health experts scored more highly for the healthy dietary pattern and had lower scores on the unhealthy dietary pattern before and during the lockdown. There was no evidence that these associations differed before and during the lockdown (*p*-value for interaction > 0.05). Conversely, participants who were influenced by nutrition and health experts had significantly higher scores in the vegan dietary pattern, however, only during the lockdown.

Participants who reported being influenced by family and friends had significantly higher scores for the healthy and the meat dietary patterns, and there was no evidence that these associations differed before and during the lockdown. Participants who reported being influenced by family and friends scored significantly lower in the vegan dietary pattern, but only before the lockdown.

Participants who reported being influenced by celebrity cooks had significantly higher scores in the meat dietary pattern before and during the lockdown, and there was no evidence that these associations differed before and during the lockdown. Individuals who reported being influenced by food influencers had significantly higher scores in the vegan dietary pattern before and during the lockdown, and there was no evidence that these associations differed before and during the lockdown (Table 3).

### 3.5. Recipe Selection and Dietary Patterns before and during the Lockdown

Table 4 presents the results of the adjusted associations between participants’ reasons for recipe selection and their dietary pattern scores before and during the lockdown.

Participants scored higher in the healthy dietary pattern before and during the lockdown when their reasons for recipes selection were ‘guaranteed to taste good’; ‘achievable with the ingredients I have at home’, ‘innovative’, ‘healthy’, and ‘environmentally friendly’. Apart from the reason ‘environmentally-friendly’ (where there were differences in the magnitude of associations before and during the lockdown), for all other reasons, there were no significant differences in the associations before and during the lockdown. Participants scored lower in the healthy dietary pattern before and during the lockdown when their reason for recipe selection was ‘easy to prepare’, with no evidence that these associations differed before and during the lockdown. Participants also scored lower in the healthy dietary pattern when their reasons for recipes selection were: ‘achievable with ingredients that can be easily found at the store’, ‘quick to prepare’, ‘comforting’ (only before the lockdown), and ‘inexpensive to make’ (only during the lockdown).

Participants scored higher in the unhealthy dietary pattern before and during the lockdown when their reasons for recipes selection were ‘easy to prepare’ and ‘comforting’, and there was no evidence that these associations differed before and during the lockdown. However, those who indicated ‘quick to prepare’ as a reason for recipe selection also scored higher in the unhealthy dietary pattern, but only before the lockdown. Participants scored lower in this dietary pattern before and during the lockdown when their reasons for recipe selection were: ‘innovative’, ‘healthy’, and ‘environmentally friendly’. Apart from the reason environmentally friendly (where there were differences in the magnitude of associations before and during the lockdown), there were no significant differences in the associations with ‘innovative’ and ‘healthy’ before and during the lockdown (Table 4).

Participants scored higher in the vegan dietary pattern before and during the lockdown when their reasons for recipes selection were ‘healthy’ and ‘environmentally friendly’, with the difference in the magnitude of these associations before and during the lockdown being only for ‘environmentally friendly’. Individuals scored lower in this dietary pattern if they indicated the following as reasons for recipe selection: ‘achievable with ingredients that can be easily found at the store’ (before the lockdown) and ‘guaranteed to taste good’ and ‘comforting’ (before and during the lockdown, with no evidence of differences in the associations in these two moments) (Table 4).

Participants scored lower in the meat dietary pattern before and during the lockdown when their reason for recipe selection was ‘inexpensive to make’, with no evidence of differences in the associations in these two moments (Table 4).

## 4. Discussion

### 4.1. Summary of Findings

We examined how changes to the sources of food- and cooking-related advice and the reasons for recipe selection related to changes in dietary patterns during the initial COVID-19 lockdown in Aotearoa New Zealand. Our results revealed that what people eat is informed by social determinants. Nutrition and health experts and family and friends were widely recognised as being significant in food- and cooking-related decisions both before and during the lockdown. Of the 2977 participants, respectively, before and during the lockdown, 50.9% and 53.9% reported being influenced by nutrition and health experts. Respectively, 35% and 29% reported being influenced by family and friends before and during the lockdown. Participants who reported being influenced by nutrition and health experts had higher scores in the healthy dietary pattern and lower scores in the unhealthy dietary pattern before and during the lockdown. ‘Taste’ was consistently reported as a major influence on recipe selection, regardless of lockdown. There were significant associations between certain food and cooking inspiration sources and dietary patterns. Participants who reported being influenced by celebrity cooks and by food influencers had, respectively, significantly higher scores in the meat dietary pattern and in the vegan dietary pattern. There were no significant differences in these associations before and during the lockdown. Finally, reasons for recipe selection were also significantly associated with dietary patterns. The most frequently reported reasons for recipe selection were ‘guaranteed to taste good’ (respectively, 88.0% and 90.9% before and during the lockdown), ‘achievable with ingredients that can be easily found at the store’ (88.0% and 88.1%), and because it is ‘healthy’ (85.5% and 80.1%). Before and during the lockdown, participants scored higher in the healthy dietary pattern when their reasons for recipes selection were ‘guaranteed to taste good’; ‘achievable with the ingredients I have at home’, ‘innovative’, ‘healthy’, and ‘environmentally friendly’. There were no significant differences in these associations before and during the lockdown.

### 4.2. Comparison of Findings in Relation to Previous Studies

As demonstrated in previous studies, there has been dramatic growth in using the Internet as a source of nutrition and dietary information since 2004; however, most adult populations still obtain their information from other sources, such as friends and family [24]. Most popular nutrition information pages online are hosted by celebrities, followed by dietitians, the weight loss industry, or other persons [4]. Half of the respondents in this study reported being influenced by nutrition and health experts for food and cooking before the lockdown, even though there is research that shows that the current online dissemination of evidence-based dietary guidelines does not have an extensive reach in the general population [4]. Nutrition and health experts as inspiration sources were found to have a significant positive relationship with healthy dietary pattern scores both before and during the lockdown and had a significant inverse relationship with unhealthy dietary pattern scores. This is expected, as nutrition and health experts, such as registered dietitians, are uniquely qualified to help consumers connect knowledge with behaviour to achieve a healthful dietary pattern [25]. However, nutrition and health experts dropped in influence during the lockdown. Many qualified scientists, registered dietitians, and other professionals have social media followings, but they may not be perceived as relatable to the consumer, or their content may not be as visually pleasing or easy to digest [26]. The least reported inspiration source included celebrity cooks in this study; however, previous studies have shown that celebrity power allows their pages to have a vast following and social media influence [4,27]. Nearly all food-related inspiration sources, except for celebrity cooks, seemingly dropped during the lockdown. With live sports on hold, movie theatres shuttered, and vacation plans scrapped in lockdown, people turned to food as a diversion, and celebrity cooks were reported as a source of comfort [28]. Nadia Lim, who is a celebrity cook with a bachelor’s degree in human nutrition and a post-graduate diploma in dietetics, appeared to influence most responders in New Zealand in this survey, because they found her to be trustworthy. Nadia Lim had a widespread reach during the lockdown, with a ‘comfort lockdown’ cooking show, which was, then, converted into a cookbook [29].

Online searches about food and shopping increased during the pandemic [30,31]. A few examples of widely used online digital tools for social connection and networking included Zoom, Microsoft Teams, and Google Hangouts [32]. Previous research has shown positive cross-sectional associations between information-seeking and healthy lifestyle behaviours such as weight management, exercise, and fruit and vegetable consumption in a general population sample [3,33]. However, providers of nutrition information online (on blogs, social media, and ‘expert’ websites) are also most commonly (34%) individuals lacking identifiable nutrition qualifications; 19% had no identifiable author information, and only 5% were from nutrition professionals, as shown in a study from 476 posts from the internet by 307 of laypeople enrolled in an open online course [5]. Research shows that an important topic of online communication about food is recipe sharing. This implies that people talk about home cooking or homemade food online, suggesting that recipe sharing and cooking homemade food still plays a big role in our culture today [34].

Household members, friends, and family noticeably influenced a high percentage of respondents in this study. Social support (i.e., assistance of family and friends) is generally associated with healthy outcomes, such as engagement in preventive health eating behaviours, and reduced smoking, alcohol and caffeine consumption [35,36]. Several studies also indicate that accommodating family members’ food preferences (i.e., partners/spouses and other family members who refuse to eat healthy foods) deterred efforts to eat healthily, particularly for women [37,38,39]. In this study, friends and family were found to have a significant inverse relationship with vegan dietary pattern scores before the lockdown and a significant positive relationship with the meat dietary pattern scores before and during the lockdown. This could be explained by the evidence in the literature that people generally view vegetarians and vegans negatively, because they severely disrupt family traditions and social conventions related to food [40,41,42,43]. The number of vegetarian/vegan friends and family may influence meat consumption, but it is unknown whether vegetarians/vegans influence their meat-eating friends and family to decrease their meat consumption [44]. Those who follow a vegetarian dietary pattern may be less likely to turn to friends and family for dietary advice, especially if the friends and family are not following a vegetarian dietary pattern. Individuals who reported being influenced by food influencers had significantly higher scores in the vegan dietary pattern. Research has found that social media food influencers provide a space that influences people to reduce their meat consumption, as vegan food influencers allow people to select knowledge on how to practice veganism that best suits their identity, beliefs, and lifestyle [45,46]. Those who choose to follow a vegetarian dietary pattern may seek out influencers on this topic. It is likely that the dietary patterns that people follow influence who they choose to be influenced by, just as much as the inspiration sources influence the dietary pattern. These findings are important, because they can be used to improve the efficacy of public health initiatives focused on encouraging plant-based diet adoption and meat consumption reduction [43].

An online survey of adult mobile device users (*n* = 583), most of whom were responsible for household grocery shopping and meal planning, found that many participants used multiple sources for meal ideas (e.g., family and friends, cookbooks, internet), which is consistent with the findings of the present study [47]. However, research examining whether recipes meet national nutritional guidelines is still in its infancy. Research has found that most recipes are nutritionally limited, and few feature a fruit or vegetable as the main ingredient [48,49]. Although healthy recipes are used to encourage home cooking and healthier eating, there is no consensus regarding best practices for types of recipes to offer or how to deliver them online most effectively to the target audience [50,51]. Research shows that cookbooks or recipes are likely to be perceived as trustworthy, particularly if the reader recognizes the author (e.g., celebrity cook Nadia Lim in this study) [23,47].

Benefits have been attributed to foods cooked at home [52], and the study participants were essentially forced to prepare four weeks of home-cooked meals during level 4 lockdown. It was expected that because of more frequent food preparation at home, people were more likely to meet dietary recommendations [53]. From a public health point of view, it is important to identify factors governing recipe choices for meal preparation, as the type of food cooked impacts the dietary patterns. Sensory appeal, health-related benefits, and meal context, as well as time and energy for food preparation, were shown to play an important role in reasons for recipe selection in this study before the lockdown and another small-scale qualitative study in the literature [54]. In the past, researchers used Twitter hashtags to track the types of foods people eat most often during the day, and the common reasons for eating were #social (activity), #taste, and #convenience [55]. Taste emerged as a significant factor in reasons for recipe selection both before and during the lockdown. Previous research in New Zealand has shown that palatability was also the most important factors for Pacific mothers when choosing food to bring home [56]. Studies that have focused on food choices identified sensory appeal as an important motive in food and, therefore, recipe selection [57,58,59,60,61,62].

Food availability (ingredients at home, easily available at store, and limited ingredients needed in the recipe) was also highly regarded as a reason for recipe selection in this study, particularly during the lockdown. Perhaps not surprisingly, given the increased inconvenience and stress created by physical distancing in supermarkets, grocery shopping enjoyment decreased during the lockdown, and some New Zealanders changed the way they shopped for food [19]. Convenience in terms of time (easy and quick to prepare) was more prevalent as a reason for recipe selection before the lockdown, and this agrees with the literature on time pressure and motives for food selection [62,63,64,65,66]. These participants searching out convenient recipes also scored lower in the healthy dietary patterns. This could be because recipes in the ‘quick and easy’ category are low sugar, but high in fat, and vegetables are not a major component [67]. However, convenience appeared to be less of a reported reason for recipe selection during the lockdown, as having more time to prepare foods meant that many experienced increased enjoyments of cooking and baking that was more time consuming in normal times, as shown by the increased baking of sourdough bread reported globally during the initial phase of the COVID-19 crisis [68]. Comforting recipes was also a prevalent reason for recipe selection among the participants in this study, particularly during the lockdown, and these participants scored higher in the unhealthy dietary pattern. The notion of ‘comforting’ foods can have positive mood benefits, especially during periods of social isolation; however, the simple carbohydrates, added sugars, high fat, and high sodium content typically found in both comfort meals and snacks can affect a person’s physical health [69].

### 4.3. Strengths and Limitations

One major limitation of the study is that the findings cannot be generalizable to the NZ adult population, as it is a convenience sample recruited online, which risks potential selection bias. In fact, the study sample resulted in a high proportion of women, highly educated participants, and most likely, people interested in food and nutrition. This could be a reason for the reported influence of nutrition and health experts for food and cooking among half of the respondents. Relative to the NZ adult population, there was low participation of Pasifika peoples, and, although Māori had a higher participation (10.6% of respondents), this was still lower than would be representative [19]. A key strength of this research, however, was the high participation in the online survey, despite the survey requiring a relatively high respondent burden (30 min to complete), a moderate literacy level, and relying on recall of respondents to assess and self-report changes in their behaviours before and during the lockdown. Given that the ‘before’ and ‘during’ reports were collected at the same time (i.e., during the lockdown), there may be bias in the reporting of behaviours, and given that there is no comparison group, the changes observed may not be fully attributable to the lockdown. The FFQ was not previously validated [70,71,72] for New Zealand’s population, because it was aimed to be used by all countries included in the international survey, and the assessment of the tool would involve a considerable amount of time. The FFQ did not collect standard servings but, instead, asked for portion by frequency of intake, which makes it challenging to assess dietary intake accurately. Since it was an international survey for multiple countries, it was also difficult to compare reported intakes with the New Zealand dietary guidelines. The survey questions and the response scales were also not validated previously, so there is potential for considerable random measurement error in responses to the questionnaire items in such a large sample size. Dichotomizing the 7-point Likert scales, while not always recommended, provided simplicity and interpretability, and the insights remain interesting.

## 5. Conclusion

### Implications for Future Research and Practice

Understanding the multiple and intersecting inspiration sources that shape dietary behaviours is the initial step in developing effective public health nutrition messaging both during and beyond a pandemic. Personalised approaches to dietary change can incorporate sources of cooking-related advice and reasons for recipe selection. These inspiration sources and motives can inform content for dietary interventions. Best-practice public health nutrition education, intervention, and recommendations can benefit from work on food- and cooking-related inspirations, influences and recipe selection motives with individuals, groups, communities, and populations [73]. The motives for food choice may also be a source of population level nutritional disparities both with lockdown and without lockdown. These findings have implications for how to communicate information about healthy and sustainable diets. Specifically, public health organisations looking to use social media to influence attitudes and behavioural intentions toward health issues (e.g., plant-based diets) should seek to reach their target audiences through selecting influencers and messages that will optimally present the health issue in a relatable and engaging way [74]. Tailoring public health messages and educational campaigns to specific food choice inspiration sources, influences, motives, and values could meaningfully inform community health initiatives and marketing strategies. Previous research has shown that due to the complexity of human food choices, the recommendation process can be manipulated through sophisticated nudging techniques such that a particular, ‘healthier’ recipe will be chosen more often than would be expected by chance alone [51]. Findings can be used to study the clustering of unhealthy dietary patterns and factors affecting diet-related disease, disease–disease interactions, and social condition interactions [73].

## Figures and Tables

**Table 1 nutrients-13-03917-t001:** Number (%) of participants according to reported food and cooking inspiration sources before and during the lockdown (*n* = 2997).

	Number (%) Influenced before the Lockdown	Number (%) Influenced during the Lockdown	Number (%) Who Became Influenced during the Lockdown	Number (%) Who Dropped Influence during the Lockdown
Family and friends	1059 (35.3)	854 (28.5)	136 (4.5)	341 (11.4)
Female (*n* = 2654)	943 (35.5)	761 (28.7)	123 (4.6)	305 (11.5)
Male (*n* = 312)	112 (35.9)	91 (29.2)	13 (4.2)	34 (10.9)
Nutrition and health experts	1521 (50.8)	1161 (38.7)	97 (3.2)	457 (15.3)
Female (*n* = 2654)	1374 (51.8)	1029 (38.8)	84 (3.2)	429 (16.2)
Male (*n* = 312)	131 (42.0)	114 (36.5)	11 (3.5)	28 (9.0)
Celebrity cooks	115 (3.8)	155 (5.2)	75 (2.5)	35 (1.2)
Female (*n* = 2654)	106 (4.0)	143 (2.8)	68 (2.6)	31 (1.2)
Male (*n* = 312)	9 (2.9)	12 (3.9)	7 (2.2)	4 (1.3)
Food influencers	304 (10.1)	292 (9.7)	92 (3.1)	104 (3.5)
Female (*n* = 2654)	276 (10.4)	269 (10.1)	86 (3.2)	93 (3.5)
Male (*n* = 312)	28 (9.0)	21 (6.7)	4 (1.3)	11 (3.5)

**Table 2 nutrients-13-03917-t002:** Number (%) of participants according to the reasons for recipe selection before and during the lockdown.

	Before the Lockdown, *n* (%)	During the Lockdown, *n* (%)
Number who actively searched for recipes/food preparation content	2063	2218
Guaranteed to taste good	1815 (88.0)	2015 (90.9)
Achievable with few ingredients	1216 (58.9)	1726 (77.8)
Achievable with the ingredients I have at home	1563 (75.8)	2065 (93.1)
Achievable with ingredients that can be easily found at the store	1816 (88.0)	1953 (88.1)
Easy to prepare	1452 (70.4)	1428 (64.4)
Quick to prepare	1415 (68.6)	1132 (51.0)
Innovative (new, something different)	1101 (53.4)	1267 (57.1)
Inexpensive to make	1205 (58.4)	1269 (57.2)
Comforting	1214 (58.9)	1682 (75.8)
Healthy	1763 (85.5)	1777 (80.1)
Environmentally friendly	1219 (59.1)	1191 (53.7)

**Table 3 nutrients-13-03917-t003:** Adjusted associations between food- and cooking-related inspiration sources and scores in the dietary patterns before and during the lockdown (*n* = 2997).

Dietary Patterns/Food and Cooking Inspiration Sources	Before the Lockdown		During the Lockdown		*p*-Value for Interaction with Lockdown
	Mean Difference (95% CI) * in Dietary Pattern Score for Those Who Were Influenced Compared to Those Who Were Not	*p*-Value	Mean Difference (95% CI) * in Dietary Pattern Score for Those Who Were Influenced Compared to Those Who Were Not	*p*-Value	
Healthy dietary pattern					
Family and friends	0.12 (0.001, 0.23)	0.047	0.13 (0.01, 0.25)	0.031	0.169
Nutrition and health experts	0.47 (0.36, 0.58)	<0.001	0.42 (0.31, 0.52)	<0.001	0.225
Celebrity cooks	−0.11 (−0.40, 0.17)	0.432	−0.18 (−0.42, 0.05)	0.129	0.522
Food influencers	0.06 (−0.12, 0.24)	0.531	0.01 (−0.17, 0.19)	0.930	0.746
Unhealthy dietary pattern					
Family and friends	0.09 (−0.02, 0.19)	0.101	0.04 (−0.07, 0.15)	0.501	0.702
Nutrition and health experts	−0.15 (−0.25, −0.05)	0.003	−0.17 (−0.27, −0.07)	0.001	0.667
Celebrity cooks	0.15 (−0.11, 0.41)	0.265	0.03 (−0.19, 0.26)	0.772	0.342
Food influencers	0.08 (−0.08, 0.25)	0.335	−0.04 (−0.21, 0.13)	0.628	0.131
Vegan dietary pattern					
Family and friends	−0.19 (−0.29, −0.09)	<0.001	−0.05 (−0.16, 0.06)	0.337	0.007
Nutrition and health experts	0.00 (−0.09, 0.10)	0.972	0.13 (0.03, 0.23)	0.009	0.001
Celebrity cooks	−0.07 (−0.32, 0.18)	0.584	0.06 (−0.16, 0.28)	0.602	0.897
Food influencers	0.23 (0.07, 0.39)	0.004	0.27 (0.10, 0.43)	0.001	0.432
Meat dietary pattern					
Family and friends	0.30 (0.20, 0.39)	<0.001	0.32 (0.22, 0.42)	<0.001	0.217
Nutrition and health experts	0.06 (−0.03, 0.15)	0.176	0.02 (−0.07, 0.12)	0.625	0.648
Celebrity cooks	0.42 (0.18, 0.66)	0.001	0.37 (0.17, 0.58)	<0.001	0.486
Food influencers	0.14 (−0.01, 0.29)	0.077	0.14 (−0.02, 0.29)	0.082	0.519

* Adjusted for gender, age, ethnicity, level of education, and household composition.

**Table 4 nutrients-13-03917-t004:** Adjusted associations between reasons for recipe selection and scores in the dietary patterns before and during the lockdown (*n* = 2664).

Dietary Patterns/Reasons for Recipes Selection	Before the Lockdown		During the Lockdown		*p*-Value for Interaction with Lockdown
	Mean Difference (95% CI) * in Dietary Pattern Score for Those Who Selected the Recipe for This Reason Compared to Those Who Did Not	*p*-Value	Mean Difference (95% CI) * in Dietary Pattern Score for Those Who Selected the Recipe for This Reason Compared to Those Who Did Not	*p*-Value	
Healthy dietary pattern					
Guaranteed to taste good	0.21 (0.05, 0.38)	0.011	0.20 (0.02, 0.38)	0.033	0.729
Achievable with few ingredients	−0.11 (−0.23, 0.01)	0.066	−0.08 (−0.21, 0.06)	0.247	0.477
Achievable with the ingredients I have at home	0.24 (0.10, 0.38)	0.001	0.27 (0.06, 0.49)	0.013	0.063
Achievable with ingredients that can be easily found at the store	−0.23 (−0.41, −0.05)	0.014	−0.03 (−0.20, 0.14)	0.705	0.657
Easy to prepare	−0.14 (−0.27, −0.01)	0.039	−0.13 (−0.24, −0.01)	0.038	0.657
Quick to prepare	−0.15 (−0.28, −0.02)	0.020	−0.07 (−0.18, 0.04)	0.222	0.728
Innovative (new, something different)	0.32 (0.21, 0.44)	<0.001	0.33 (0.22, 0.45)	<0.001	0.218
Inexpensive to make	0.08 (−0.03, 0.20)	0.164	−0.12 (−0.23, −0.004)	0.043	0.011
Comforting	−0.20 (−0.32, −0.09)	0.001	−0.11 (−0.24, 0.02)	0.091	0.457
Healthy	1.02 (0.87, 1.17)	<0.001	0.88 (0.74, 1.01)	<0.001	0.111
Environmentally friendly	0.86 (0.75, 0.97)	<0.001	0.72 (0.61, 0.83)	<0.001	0.005
Unhealthy dietary pattern					
Guaranteed to taste good	−0.02 (−0.18, 0.13)	0.754	0.00 (−0.17, 0.17)	0.980	0.583
Achievable with few ingredients	0.10 (−0.01, 0.21)	0.065	0.10 (−0.03, 0.22)	0.140	0.295
Achievable with the ingredients I have at home	−0.05 (−0.17, 0.08)	0.435	0.12 (−0.08, 0.32)	0.244	0.017
Achievable with ingredients that can be easily found at the store	0.16 (−0.001, 0.33)	0.052	0.02 (−0.14, 0.18)	0.817	0.231
Easy to prepare	0.21 (0.09, 0.33)	0.001	0.22 (0.11, 0.33)	<0.001	0.705
Quick to prepare	0.14 (0.02, 0.26)	0.017	0.10 (−0.002, 0.21)	0.054	0.178
Innovative (new, something different)	−0.25 (−0.36, −0.15)	<0.001	−0.27 (−0.38, −0.17)	<0.001	0.470
Inexpensive to make	−0.09 (−0.20, 0.02)	0.118	−0.05 (−0.15, 0.06)	0.410	0.562
Comforting	0.49 (0.38, 0.59)	<0.001	0.50 (0.38, 0.62)	<0.001	0.557
Healthy	−0.66 (−0.81, −0.52)	<0.001	−0.69 (−0.82, −0.57)	<0.001	0.468
Environmentally friendly	−0.38 (−0.49, −0.27)	<0.001	−0.39 (−0.49, −0.29)	<0.001	0.613
Vegan dietary pattern					
Guaranteed to taste good	−0.16 (−0.31, −0.01)	0.032	−0.22 (−0.39, −0.05)	0.010	0.272
Achievable with few ingredients	0.03 (−0.07, 0.14)	0.518	−0.08 (−0.21, 0.04)	0.183	0.143
Achievable with the ingredients I have at home	−0.03 (−0.15, 0.09)	0.612	0.05 (−0.15, 0.24)	0.657	0.168
Achievable with ingredients that can be easily found at the store	−0.18 (−0.34, −0.02)	0.026	0.07 (−0.08, 0.23)	0.360	0.284
Easy to prepare	−0.07 (−0.18, 0.05)	0.257	−0.08 (−0.19, 0.03)	0.145	0.303
Quick to prepare	−0.08 (−0.19, 0.04)	0.194	0.01 (−0.09, 0.12)	0.822	0.552
Innovative (new, something different)	−0.02 (−0.12, 0.09)	0.730	0.10 (−0.01, 0.20)	0.065	0.002
Inexpensive to make	0.03 (−0.08, 0.13)	0.600	0.07 (−0.03, 0.18)	0.165	0.826
Comforting	−0.13 (−0.23, −0.02)	0.017	−0.16 (−0.28, −0.04)	0.008	0.595
Healthy	0.23 (0.09, 0.37)	0.001	0.38 (0.26, 0.51)	<0.001	<0.001
Environmentally friendly	0.43 (0.33, 0.53)	<0.001	0.58 (0.48, 0.68)	<0.001	<0.001
Meat dietary pattern					
Guaranteed to taste good	−0.01 (−0.14, 0.13)	0.935	0.16 (0.003, 0.32)	0.046	0.697
Achievable with few ingredients	−0.01 (−0.11, 0.09)	0.827	0.02 (−0.10, 0.13)	0.790	0.713
Achievable with the ingredients I have at home	−0.10 (−0.21, 0.02)	0.098	−0.09 (−0.27, 0.10)	0.366	0.353
Achievable with ingredients that can be easily found at the store	0.02 (−0.13, 0.17)	0.768	−0.05 (−0.20, 0.10)	0.499	0.821
Easy to prepare	−0.03 (−0.14, 0.08)	0.582	0.03 (−0.08, 0.13)	0.617	0.233
Quick to prepare	−0.01 (−0.11, 0.10)	0.922	−0.01 (−0.11, 0.09)	0.815	0.678
Innovative (new, something different)	0.08 (−0.02, 0.18)	0.099	0.06 (−0.04, 0.15)	0.260	0.220
Inexpensive to make	−0.20 (−0.30, −0.10)	<0.001	−0.17 (−0.27, −0.07)	0.001	0.515
Comforting	0.00 (−0.10, 0.10)	0.964	0.04 (−0.07, 0.15)	0.462	0.143
Healthy	−0.08 (−0.22, 0.05)	0.215	−0.03 (−0.15, 0.09)	0.666	0.076
Environmentally friendly	−0.35 (−0.44, −0.25)	<0.001	−0.36 (−0.46, −0.27)	<0.001	0.025

* Adjusted for gender, age, ethnicity, level of education, and household composition.

## Data Availability

Data are available from the authors on request.

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
