# Peer review of "Who We Seek and What We Eat? Sources of Food Choice Inspirations and Their Associations with Adult Dietary Patterns before and during the COVID-19 Lockdown in New Zealand"

_nutrients, 2021, doi:10.3390/nu13113917_

Round 1

Reviewer 1 Report

Firstly:

1. The information about affiliation is untidy (different spacing from numeration[e.g. line 9,10], difference e-mails format [e.g. 17,18, 9,10 ….] and no information who is corresponding author in line [line 5-7, 19].  

2. Proposed keywords are not aligned with Pubmed MESH subheadings. New Zeland it is not good key word. 

3. Line 42 and 44 - the same citation - if you used only no 1 you can use after second sentence.

4. The authors should explain whether their study sample is representative of the New Zeland population concerning all relevant socio-demographic and socio-economic characteristics. In my opinion, gender where you have 88,6% of females, and in the general population, it is not even close to this percentage.

5. Please describe exactly the inclusion and exclusion criteria

6.  The authors should clearly state that this simple is a non-representative sample due to gender, regional, and other socio-demographic factors. Also, the authors have a huge bias because they organized online surveys, and automatically excluded those without the internet or older people who usually use the internet rarely. This should be stated also in the limitations section.

7. In Nutrients Microsoft Word templat on page one you have how the abstract must look: Abstract: Background, Methods, Results, Conclusions. This section should be rewriten. 

8. When you will create "Conclusions section" in your manuscript, it will be more readable.  

Reviewer 2 Report

The submitted manuscript is very timely, as it analyzes the impact of sources of inspiration and their relationship with eating patterns. However, some important comments need to be addressed.

Abstract

- I suggest the authors to rephrase the objective since it does not read well.

- This part of the manuscript should indicate the study design and give some details about the statistical analyzes that were used to analyze the data.

- In addition, the results should be presented in a less descriptive way.

Introduction:

  • Lines 107-109: Is this your hypothesis? If so, please indicate so.
  • Lines 109 – 114: I think that the authors are mixing objectives with methodology. Please, indicate only the objectives of the research

Materials and Methods:

  • Please simplify point 2.2, it is really difficult to follow. Perhaps a table would help the reader to understand the questions and the possible answers. I had a hard time conceptualizing the questionnaire.
  • Point 2.3 should also be organized. It seems that you are mixing the methodology with results. I highly recommend describing the results obtained from the PCA in the results section.
  • Line 97: can you clarify what type of food was included in the “un processed vegetarian alternatives”

Statistical analysis:

  • Some details about the PCA analysis are appropriate in this section
  • I strongly recommend adjusting p-values for multiple comparisons

Results:

  • Please include the PCA results in this section.
  • Have you collected anthropometric data? I think it would be interesting for the readers to know the impact of their choices and influences on health outcomes (e.g. BMI).

Discussion section:

  • I suggest that the authors shorten the discussion. I also recommend making it a little lighter to read. I think the results are interesting, however, as it is very difficult to follow.

Table captions:

  • Captions should be self-explanatory. Please add details about statistical analyses that were used.

Supplementary Figure 1

  • Note that the study design is cross-sectional, so the STROBE guidelines, rather than CONSORT, would be more appropriate.
